# The Fatigue-Related Symptoms Post-Acute SARS-CoV-2: A Preliminary Comparative Study

**DOI:** 10.3390/ijerph191811662

**Published:** 2022-09-16

**Authors:** Marie Thomas

**Affiliations:** Reader in Psychology, Centre for Health and Cognition, Bath Spa University, Bath BA2 9BN, UK; m.thomas4@bathspa.ac.uk

**Keywords:** post-acute SARS-CoV-2, fatigue-related symptoms, chronic fatigue syndrome, non-fatigued controls

## Abstract

A sizeable sub-group of individuals continue to experience persistent debilitating symptoms post-acute SARS-CoV-2. Although these can vary from person to person, fatigue appears to be the most common symptom. Post-viral fatigue has been documented in conditions such as influenza, infectious mononucleosis and more recently chronic fatigue syndrome (CFS). The current study uses measures that successfully describe the fatigue-related symptoms associated with CFS to investigate the fatigue experienced post-acute SARS-CoV-2. Twenty-six volunteers were recruited from Long COVID support groups active on social media. Data were collected anonymously using an online survey platform. These data were compared to pre-pandemic data from non-fatigued and CFS groups. The post-acute SARS-CoV-2 volunteers reported significantly higher levels of fatigue and cognitive difficulties than the non-fatigued controls. They also report more individual symptoms (such as lack of concentration) and problems with sleep quality. There was a similarity between the post-acute SARS-CoV-2 volunteers and the CFS group in terms of levels of depression, perceived stress, emotional distress and cognitive difficulties. Although this was a small-scale study, it demonstrates the range of symptoms experienced post-acute SARS-CoV-2. In addition, the similarities between this group and CFS suggests the need for further research into the mechanisms at play here, the need to identify those at risk of long-term symptoms and the development of possible interventions.

## 1. Introduction

The symptoms that present during the acute phase of SARS-CoV-2 vary in severity and duration. What is becoming increasingly clear is that, for some, symptoms can persist after the initial infectious phase [1], referred to as Long COVID [2]. It is not clear why a subgroup of individuals goes on to develop Long COVID and what risk factors predispose people to it. It is also not clear if these symptoms will resolve over time or require intervention. One of the most frequently reported symptoms associated with Long COVID is fatigue [1,2].

Fatigue is the expected result of over-exertion or lack of sleep and has been defined as an intense subjective sense of tiredness, energy depletion and weakness. The subjective nature of fatigue means that it is interpreted differently by different individuals. It manifests both physical (e.g., muscle weakness) and cognitive (e.g., reduced levels of concentration) symptoms and can range from tiredness to clinically relevant exhaustion. Taken together, these aspects of fatigue make it difficult to quantify accurately for the purpose of research [3].

We have already described the range of cognitive impairments resulting from experimentally induced acute fatigue [4]. We have also charted the effect of the common cold and influenza on mood and cognitive performance [5,6]. The after-effects of colds and influenza, which can last for weeks, have also been discussed [7,8]. In the case of infectious mononucleosis, effects can be reported for months post-infection [9]. The Epstein–Barr virus was also implicated as a causative agent in Post Viral Fatigue Syndrome (PVFS)—a term once used to describe chronic fatigue syndrome (CFS) [10]. 

CFS provides us with evidence of how debilitating persistent fatigue that is unresolved by rest can be. Those with the condition experience substantial measurable cognitive impairment together with high levels of somatic symptoms, anxiety and depression [3]. The decreased personal, occupational and social activities which impact on their quality of life result in individuals with CFS being more likely to be unemployed than their peers [11]. 

We are now seeing increased reporting of long-term fatigue following the acute stage of SARS-CoV-2. Data from the Office for National Statistics (ONS) estimate that 1.8 million (2.8% of the population) in the UK are reporting symptoms more than 4 weeks post-acute infection (ONS.gov.uk, 2022). In a recent paper, Smith [12] discusses the implications of Long COVID and calls for research into the risk factors and mechanisms underlying the condition. To address the need to describe the fatigue-related symptoms associated with Long COVID (LC), this preliminary study employs the subjective measures used in other chronic conditions where significant fatigue is reported [13,14].

## 2. Materials and Methods

A cross-sectional online survey design methodology was employed.

### 2.1. Participants

#### 2.1.1. Post-Acute SARS-CoV-2

Adults who had been infected with the SARS-CoV-2 virus were recruited from Long COVID support groups via Twitter^TM^. Inclusion criteria for the study were: (a) adults over the age of 18 years who believed that they had been infected with SARS-CoV-2 (COVID-19) who (b) were experiencing ongoing symptoms (Long COVID). Exclusion criteria for the study were: (a) anyone under 18 years of age who did not believed that they had been infected with SARS-CoV-2 (COVID-19); (b) were not experiencing ongoing symptoms (Long COVID); (c) people who lack mental capacity, may be at risk of losing capacity or have fluctuating capacity; (d) people who suffer from psychiatric or personality disorders, including those conditions in which capacity to consent may fluctuate; (e) people who may have only a basic or elementary knowledge of the English language; (f) people who may socially not be in a position to exercise unfettered informed consent; and (g) any members of staff or students working at the host university.

A total of 31 adults began the questionnaire. Data were excluded if all aspects of the survey were not completed. Of the 29 remaining cases, 3 were excluded from the analysis as they were the only participants who had been hospitalised due to the infection. This precluded them from any meaningful comparative analysis.

#### 2.1.2. Non-Fatigue Controls

Data from non-fatigue individuals were used as the first comparison group. These anonymised data were collected before the pandemic from volunteers recruited from university staff [14]. Inclusion criteria for the study were: (a) adults over the age of 18 years who (b) were not experiencing symptoms of chronic fatigue. Exclusion criteria for the study were: (a) anyone under 18 years of age who did not believed that they had CFS; (b) people who lack mental capacity, may be at risk of losing capacity or have fluctuating capacity; (c) people who suffer from psychiatric or personality disorders, including those conditions in which capacity to consent may fluctuate; (d) people who may have only a basic or elementary knowledge of the English language; and (e) people who may socially not be in a position to exercise unfettered informed consent. Participants from the study consented to their data being used in research articles.

#### 2.1.3. Chronic Fatigue Syndrome (CFS)

Data from individuals with CFS were used as the second comparison group. These anonymised data were collected before the pandemic from volunteers recruited from the Action for ME support group website [14]. Inclusion criteria for the study were: (a) adults over the age of 18 years who (b) believed they currently had CFS. Exclusion criteria for the study were: (a) anyone under 18 years of age who believed that they had CFS; (b) people who lack mental capacity, may be at risk of losing capacity or have fluctuating capacity; (c) people who suffer from psychiatric or personality disorders, including those conditions in which capacity to consent may fluctuate; (d) people who may have only a basic or elementary knowledge of the English language; and (e) people who may socially not be in a position to exercise unfettered informed consent. Participants from the study consented to their data being used in research articles. 

### 2.2. Procedures

The questionnaires were accessed via the onlinesurvey.ac.uk data collection tool. Prior to completing the survey, individuals were informed about the purpose of the study and the requirements for participation. Participants were assured their answers would remain confidential and of their right to exit the questionnaire at any point. Access to the questionnaires was only granted once participants had provided dated informed consent at the beginning of the survey. Data were not collected that could identify individuals.

### 2.3. Measures

Participants provided the following demographic information: age, sex (male, female, non-binary, prefer not to say) and marital status (single, married/partner, divorced/separated, widowed). This was followed by a number of questions about (a) vaccination status and (b) SARS-CoV-2 test and date. A list of underlying conditions associated with post- SARS-CoV-2 infection was also administered [1] together with a quality of health measure [15]. This was followed by established self-report measures used to describe the symptoms associated with chronic fatigue [13,14]. 

#### 2.3.1. Anxiety

The State-Trait Anxiety Inventory [16] is a 20-item scale that relates to the person’s general propensity to experience anxiety (trait anxiety) on a 5-point Likert scale from 0 (not at all) to 4 (very much so). Higher scores indicate greater levels of anxiety.

#### 2.3.2. Depression

The Centre for Epidemiological Studies Depression (CES-D) Scale [17] is a 20-item scale that was developed to measure symptoms of depression in the general population. The frequency of individual symptoms is reported in the last week using a 5-point Likert scale ranging from 0 (rarely or none of the time) to 4 (most or all of the time). Higher scores indicate greater levels of depression. 

#### 2.3.3. Positive and Negative Affect

This 30-item scale produces measures for both positive and negative affect [18]. Participants are asked to rate how they have felt in the past week on a 5-point Likert scale ranging from 0 (not at all) to 4 (extremely). High scores on the positive affect sub-scale indicate greater positive mood, whereas high scores on the negative affect sub-scale indicate a more negative mood. 

#### 2.3.4. Self-Esteem

This measure was designed to assess factors of social confidence, ability, and self-regard [19]. Fourteen items or situations were posed to participants using a 6-point scale from 1 (agree very much) to 6 (disagree very much). Higher scores indicate greater levels of self-esteem.

#### 2.3.5. Stress

The perceived stress scale (PSS) [20] assesses the extent to which a person views situations in their life as stressful, uncontrollable or overloading. The measure asks participants to rate how often they have felt or thought a certain way during the past month on a 5-point Likert scale ranging from 0 (never) to 4 (very often). Higher scores indicate greater levels of perceived stress. 

#### 2.3.6. Cognitive Failures

The Cognitive Failures Questionnaire (CFQ) [21] is a 25-item measure designed to assess failures in perception, memory and motor function over the past six months on a 5-item scale ranging from 0 (never) to 4 (very often). Higher scores are indicative of greater cognitive failures. 

#### 2.3.7. Fatigue Related Symptoms

The profile of fatigue-related symptoms (PFRS) [22] is a 54-item measure designed specifically to assess fatigue and related symptoms in those with chronic fatigue syndrome. Each item consists of statements pertaining to the past week on a 7-item Likert scale ranging from 1 (not at all) to 7 (extremely). The PFRS consists of four subscales: emotional distress, fatigue, cognitive difficulty and somatic symptoms. Higher scores indicate greater levels of symptoms. 

#### 2.3.8. Individual Symptoms

The symptoms checklist [23] was developed to assess the level of individual symptoms reported in chronic fatigue syndrome. Twenty-seven physical and psychological symptoms are presented for individuals to select if they currently experience them. The higher the total score, the higher the symptomology.

#### 2.3.9. Quality of Sleep

The sleep behaviour scale [13] asks individuals about how many hours they sleep per night together with three questions regarding the quality of their sleep on a 5-item Likert scale ranging from 0 (never) to 4 (very often).

### 2.4. Analysis

Data analysis was conducted using Statistical Package for Social Sciences (SPSS v27). The SARS-CoV-2 data were compared to an existing anonymised dataset of non-fatigued controls and participants diagnosed with chronic fatigue syndrome (CFS). Multivariable analysis of variance was used to compare the fatigue-related measures of the SARS-CoV-2 and non-fatigued control groups and the SARS-CoV-2 and CFS groups. The significant group differences between the CFS and non-fatigued control groups across all questionnaires have been described previously [14]. 

### 2.5. Ethical Considerations

Ethical approval for the study was granted by the university Research Ethics Committee (25 November 2020). Participants gave informed consent to allow their anonymised data to be used in journal articles and conferences.

## 3. Results

### 3.1. Group Characteristics

Table 1 describes the demographic data for the three comparison groups.

#### 3.1.1. Post-Acute SARS-CoV-2 (PAC-19)

Data were collected between March and August 2021 from individuals identifying as experiencing symptoms of Long COVID (such as fatigue). The group comprised 4 males and 22 females with a mean age of 46 years (SD 2.69). Of the participants, 9 were single, 15 married and 2 divorced. A total of 18 had received at least one dose of the SARS-CoV-2 vaccine, 18 were regularly active and all were non-smokers. Of the 11 underlying conditions described by Carfi et al. [1], 1 participant reported chronic heart disease, 2 hypertension, 1 diabetes, thyroid disease and 1 had chronic obstructive pulmonary disease.

Of the 26 respondents, SARS-CoV-2 had been formally diagnosed in 17 cases: 12 by antigen test and 5 by antibody tests. One respondent had received both forms of test. The remaining participants did not respond to the question. A total of 9 participants reported symptoms of high temperature, 8 reported a new continuous cough, 9 had loss or change of in sense of taste or smell and 5 had all three of the above symptoms. None of the volunteers were asymptomatic and none of them were hospitalised.

In total, 18 participants reported receiving a first dose of a SARS-CoV-2 vaccine and 8 had not received a vaccine. Seven received the AstraZenica/Oxford vaccine and eleven received the Pitzer-BioNTech vaccine. Vaccination dates were between December 2020 and June 2021. Eleven participants had received a second vaccine dose. Two had received the AstraZenica/Oxford vaccine and nine the Pitzer-BioNTech vaccine. Second vaccines were received between February 2021 and July 2021. In total, 11 participants had received 2 vaccine doses.

When asked if SARS-CoV-2 had worsened their quality of life, 25 individuals said that it had. The quality of health scale [15] data ranged from 15 to 100 and elicited that the group reported a mean score of 52.69 (SD 27.73). The time that had elapsed since their diagnosis and completing the survey averaged 41.4 (SD 18.71) weeks (range 1–66).

#### 3.1.2. Non-Fatigue Controls

Data from 45 participants acted as the first comparison group. This group comprised 12 males and 33 females, with a mean age of 34 years (SD 1.65). Thirty of the participants were single and fifteen married.

#### 3.1.3. Chronic Fatigue Syndrome (CFS)

Data from 76 participants acted as the second comparison group. This group comprised 9 males and 68 females, with a mean age of 44 years (SD 1.52). One participant did not wish to indicate their sexual orientation. Of the participants, 35 were single, 36 married and 6 divorced. The time that had elapsed since their CFS diagnosis and completing the survey averaged 7.4 years (range from less than 1 year to 30 years).

### 3.2. Comparisons between Post-Acute SARS-CoV-2 (PAC-19) and Controls

There was no difference between the two groups in terms of male: female ratio. PAC-19 group was, however, significantly older (F(1, 60) = 13.89; *p* < 0.001) and less likely to be single (C^2^ = 8.86 df = 1 *p* = 0.012).

#### 3.2.1. Sleep Quality

There was no difference between the average number of hours slept per night or in terms of feeling rested by sleep. However, the PAC-19 group were significantly more likely to report difficulty falling asleep (C^2^ = 9.47 df = 1 *p* = 0.05) and waking early from sleep (C^2^ = 35.33 df = 1 *p* < 0.001).

#### 3.2.2. Fatigue and Related Symptoms

Table 2 describes the mean derived scores of the questionnaire measures of the post-acute SARS-CoV-2 (PAC-19) and non-fatigued control sample.

Table 2 indicates that there was no difference between the two groups in terms of anxiety, self-esteem, perceived stress or emotional distress. The PAC-19 group were significantly more likely to be depressed (F(1, 60) = 4.03; *p* = 0.049) and had significantly higher negative affect scores (F(1, 60) = 30.05; *p* < 0.001). Interestingly, they were also more likely to report higher levels of positive affect (F(1, 60) = 3.99; *p* = 0.050) than the controls. 

The PAC-19 group experienced more somatic symptoms (F(1, 60)= 3.99; *p* = 0.050)) and more individual symptoms (F(1, 60) = 26.21; *p* < 0.001) than the controls. There was significant cognitive impairment in the PAC-19 group on the PFRS (F(1, 60) = 35.01; *p* < 0.001). Although this was not corroborated by the cognitive failures questionnaire, the difference appears marginally significant (F(1. 60) = 3.75; *p* = 0.057). 

As expected, the PAC-19 group reported significantly higher levels of fatigue than the controls (F(1, 60) = 29.83; *p* < 0.001). 

### 3.3. Comparisons between Post-Acute SARS-CoV-2 (PAC-19) and CFS

There was no difference between the two groups in terms of male to female ratio, marital status or age.

#### 3.3.1. Sleep Quality

There was no difference between the two groups in terms of average number of hours slept per night or in terms of difficulty falling asleep at night. However, the PAC-19 were significantly more likely to be rested by sleep (C^2^ = 49.01; df = 4 *p* < 0.001) and more likely to wake early from sleep (C^2^ = 11.90; df = 4; *p* = 0.018) than the CFS group.

#### 3.3.2. Fatigue Related Symptoms

Table 3 describes the mean derived scores of the questionnaire measures of the post-acute SARS-CoV-2 (PAC-19) and chronic fatigue syndrome (CFS) sample.

Overall, these data indicate a mixed picture. There was no significance between the PAC-19 and CFS groups in terms of the levels of depression, perceived stress, self-esteem and emotional distress and cognitive difficulties on the profile of fatigue related symptoms scale.

However, the PAC-19 group report significantly lower levels of anxiety (F(1, 92) = 6.49; *p* = 0.013), fatigue (F(1, 92) = 8.86; *p* = 0.004), somatic symptoms (F(1, 92) = 9.05; *p* = 0.003), individual symptoms (F(1, 92) = 23.69; *p* < 0.001) and cognitive impairment on the cognitive failures questionnaire than the CFS group (F(1, 92) = 4.34; *p* = 0.040).

## 4. Discussion

The increased reporting of chronic fatigue post-acute SARS-CoV-2 presents society with a serious problem. The pervasive nature of the cognitive impairments associated with long-term fatigue has been documented in chronic fatigue syndrome (CFS) [13]. The impairments reported subjectively by individuals with CFS were corroborated by measures of objective performance and were later used to evaluate interventions [24,25]. If the persistence and level of impairment seen in CFS are being mirrored by individuals post-acute SARS-CoV-2, it will impact negatively on all aspects of their lives. This includes increased comorbid psychopathology, decreased social engagement and inability to continue gainful employment [11]. 

Importantly, this study assessed the level of impairment in post-acute SARS-CoV-2 participants using measures known to be sensitive to fatigue, and so contributes a unique perspective to the expanding literature on Long COVID. These measures have been validated in studies carried out over the previous three decades. This allows for a more accurate assessment of clinical impairment as responses can be compared to known benchmarks. 

The current study aimed to describe post-acute SARS-CoV-2 fatigue by comparison with non-fatigued controls. CFS data were used as a benchmark of the pervasive nature of long-term fatigue on quality of life. Both comparison group data were collected before the pandemic using a similar online survey method [14].

Initial comparisons between the post-acute SARS-CoV-2 group (PAC-19) and the non-fatigued controls provides evidence of fatigue related impairments. They are showing signs of poor sleep quality regardless of the number of hours sleep. The PAC-19 group reported higher levels of fatigue, depression, somatic and total symptoms. Higher levels of cognitive difficulties were reported on the profile of fatigue related symptoms (PFRS) [22] but were not corroborated by the cognitive failures questionnaire [21].

Compared to the CFS data, there are some similarities regarding the impairments being reported. The PAC-19 group reported difficulty falling asleep but were more likely to feel rested by sleep and awaken early. They also reported levels of depression, stress, emotional distress and cognitive difficulties that are similar to the CFS group. Although the levels of impairment reported in the post-acute SARS-CoV-2 group are not as wide-ranging as CFS, it is judged that these findings should be viewed with caution. This is due to the time that had elapsed between diagnosis and the data collection point, which is measured in weeks for PAC-19 and years for the CFS group. We already know that the longer fatigue is allowed to continue without intervention, the more entrenched the cognitive, behavioural, emotional, physiological and social factors become [26]. 

### 4.1. Limitations

The range of measures in the current study were chosen because they have high internal reliability and test-retest reliability. However, the sample size is small and confined to individuals who did not receive any inpatient treatment, indicating that the acute infection was relatively mild. Although we included questions around underlying conditions, there was insufficient data to act as a comparison. Similarly, data were not collected for ethnicity and socioeconomic status. There is also the need to recruit a non-fatigued post-acute SARS-CoV-2 comparison group and individuals who do think that they have been infected by SARS-CoV-2.

### 4.2. Conclusions

The current study offers a preliminary insight into the nature of the impairment reported by a number of cases following the post-acute phase of SARS-CoV-2, particularly involving fatigue related symptoms. As expected, these data highlight areas of concern regarding increased impairment reported by this group when compared to non-fatigued controls. Comparisons with CFS, however, present a less clear picture. However, differences between the time elapsed post-acute phase and the length of illness data of the CFS group raises cause for concern. When fatigue is persistent, the impairments can become intrenched.

### 4.3. Future Research

To address the sample size limitations outlined above, NHS approval has now been granted (IRAS project no. 303626). Long COVID outpatient clinics across the UK will be invited to advertise a new version of the survey. Additional ethnicity and socioeconomic data will be collected along with underlying conditions, acute phase severity and treatment and lifestyle factors to identify at risk groups.

### 4.4. Recommendations

This study concurs with the recommendations by Smith [12]. We now need to fully investigate underlying risk factors and mechanisms and consider how we might develop appropriate prevention and management strategies.

## Figures and Tables

**Table 1 ijerph-19-11662-t001:** The age, sex and marital status demographic data for the SARS-CoV-2 (PAC-19), non-fatigue controls and the chronic fatigue syndrome (CFS) groups.

	PAC-19 (*n* = 26)	Controls (*n* = 45)	CFS (*n* = 76)
Age in years (SD)	46 (2.69)	34 (1.65)	44 (1.52)
Sex: Male:Female	4:22	12:33	9:68
Marital Status %:			
Single	9	30	35
Married/Partner	15	15	36
Divorced/Separated	2	0	6
Widowed	0	0	0

**Table 2 ijerph-19-11662-t002:** The mean derived scores for the fatigue related measures. Higher scores indicated greater levels of impairment for each measure. Standard deviation (SD) scores are in parenthesis.

	PAC-19	Controls	F(1, 60), *p*
Anxiety	46.53 (2.89)	40.71 (1.78)	ns
Depression	42.47 (3.04)	35.31 (1.87)	4.03, 0.049
Positive affect Negative affect	39.88 (3.25) 35.59 (3.01)	32.27 (1.99) 16.22 (1.85)	3.99, 0.050 30.05, 0.001
Self-Esteem	56.00 (3.96)	56.91 (2.44)	ns
Perceived Stress	27.23 (2.52)	24.29 (1.55)	ns
Cognitive Failures	51.32 (3.70)	42.38 (2.76)	ns
Profile of Fatigue Related Symptoms:			
Fatigue	57.12 (4.47)	28.47 (2.75)	29.83, 0.001
Emotional distress	45.59 (5.56)	37.16 (3.42)	ns
Cognitive difficulties	43.35 (3.80)	24.13 (2.34)	35.01, 0.001
Somatic symptoms	46.71 (4.53)	30.38 (2.79)	3.99, 0.050
No. of individual symptoms	9.23 (1.05)	2.93 (0.64)	26.21, 0.001

**Table 3 ijerph-19-11662-t003:** The mean scores for a range of fatigue-related measures. Higher scores indicated greater levels of impairment for each measure. Standard deviation (SD) scores are in parentheses.

	PAC-19	CFS	F(1. 92), *p*
Anxiety	46.53 (2.89)	54.74 (1.37)	6.49, 0.013
Depression	42.47 (3.04)	48.84 (1.54)	ns
Positive affect Negative affect	39.88 (3.25) 35.59 (3.01)	23.69 (1.32) 28.62 (1.44)	3.99, 0.050 30.05, 0.001
Self-Esteem	56.00 (3.96)	45.22 (2.09)	ns
Perceived Stress	27.23 (2.52)	32.47 (1.18)	ns
Cognitive Failures	51.32 (3.70)	59.74 (2.04)	4.34, 0.040
Profile of Fatigue Related Symptoms:			
Fatigue	57.12 (4.47)	69.82 (1.81)	8.86, 0.004
Emotional distress	45.59 (5.56)	57.69 (2.84)	ns
Cognitive difficulties	43.35 (3.80)	51.32 (1.90)	ns
Somatic symptoms	46.71 (4.53)	64.17 (2.47)	9.05, 0.003
No. of individual symptoms	9.23 (1.05)	16.69 (0.65)	23.69, 0.001

## Data Availability

The data presented in this study are available on request from the corresponding author.

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
