# Peer review of "The Fatigue-Related Symptoms Post-Acute SARS-CoV-2: A Preliminary Comparative Study"

_ijerph, 2022, doi:10.3390/ijerph191811662_

Round 1
Reviewer 1 Report
I read the article titled "The fatigue-related symptoms post-acute SARS-2-CoV-2: 2 a preliminary comparative study" with great interest. Here are some points that need to be addressed by the author
1- There are many sentences in the introduction section without citation to the reference.
2- The author should recheck the abbreviations and spell them out for the first time then use the abbreviation in the rest of the context.
3- The aim of the study and the methodology of the research is not clear. The author should improve them, for example, the exact number of each group and the details of them (who is the control group and how many participants)?. I would propose to make a table to address all demographic characteristics of the participants.
4- The presentation of the results should be in the tables and the latter should not only involve mean(SD), but it should be inclusive of p-value and F.
5. The discussion should focus on the exact results, from the most important to the least.
6. There should be a conclusion section to show whether the study achieves the aim of the study or not.
Author Response
Thank you reviewing the manuscript. I have addressed your comments as follows:
- There are many sentences in the introduction section without citation to the reference.
This has been rectified
- The author should recheck the abbreviations and spell them out for the first time then use the abbreviation in the rest of the context.
This has been rectified
- The aim of the study and the methodology of the research is not clear. The author should improve them, for example, the exact number of each group and the details of them (who is the control group and how many participants)?. I would propose to make a table to address all demographic characteristics of the participants.
A clearer report of the comparison groups has been added in the method section and a demographic data table added.
- The presentation of the results should be in the tables and the latter should not only involve mean (SD), but it should be inclusive of p-value and F.
The p-values and F have been added to the tables.
- The discussion should focus on the exact results, from the most important to the least.
This has been addressed.
- There should be a conclusion section to show whether the study achieves the aim of the study or not.
Conclusion section has been added.
Reviewer 2 Report
Dear Editor,
Thank you for the opportunity to review this interesting article.
The Author present an original article that aims to investigate the fatigue experienced by patients after acute COVID-19. However, I have a few comments:
1. In the abstract, lines 7, 11, 14, 17, 19, 23: "SAR-2-CoV-2" please correct to "SARS-CoV-2".
2. In the abstract (line 11) and part of the results (line 151), the Author state that the analysis was performed on 26 patients. But then they write about a group of 25 patients (line 157). How many patients were in the 25 or 26 group? Please address this inconsistency.
3. Similarly, in the "Results" section (subsection 3.1.3) lines 172-173. The Author write that the CFS group had a total of 76 patients, but later write that there were 9 men and 68 women, which gives a total of 77 patients - please explain.
4. In the section "Introduction" (line 32) the Author write "One of the most frequently reported symptoms associated with Long Covid is fatigue." - what is the source of this information? there is no literature.
5. In the "Introduction" section, the Author write "Fatigue is the expected result of over-exertion or lack of sleep and has been defined as an intense subjective sense of tiredness, energy depletion and weakness." - what is the source of this definition? there is no literature.
6. Is it possible to specify the time period from which the study is? - did all patients meet the WHO definition of long COVID (persistence of symptoms> 12 weeks)?
7. The Author write: "Of the 25 respondents, SARS-2-CoV-2 had been formally diagnosed in 17 cases. Twelve by antigen test and 5 by antibody tests." - and in other cases, on what basis was the diagnosis of COVID-19 made?
8. There is no detailed description of the criteria that the subjects had to meet in order to be included in the study - to the PAC19 group, and what were the exclusion criteria, e.g. age above ... or below ... etc.
9. Similarly, there is a lack of precise information regarding the inclusion and exclusion criteria for the non-fatigue controls group and the CFS group.
10. It is not clear whether the "non-fatigue controls" group are people who have contracted COVID-19 who have not noticed fatigue, or people who have not suffered from COVID-19 at all.
11. Information on marital status was collected from patients - was there a relationship between the occurrence of fatigue and the marital status of patients?
12. There is no information anywhere whether the studied patients were taking any medications, eg due to the presence of comorbidities.
13. There is also no information about whether they took any medications during the acute phase of COVID-19 - please add this information as it may be important in the persistence of the observed symptoms.
14. In the "Introduction", the Author write: "We have already described the range of cognitive impairments resulting from experimentally induced acute fatigue [3]. This study also highlighted how caffeine can mitigate these effects." - the study collected information related to the lifestyle of patients (diet), e.g. did the respondents use / use caffeinated products? E.t.c.
15. In the study, in the "Material and methods" section, the Author write that patients were asked about their vaccination status - but there is no reference to the data obtained - whether in patients after vaccination, fatigue was more or less frequently observed, or whether it was not observed at all. impact? do vaccinations increase / decrease the risk of fatigue after COVID-19?
16. How many patients in this group have completed the full vaccination course? How many was 1? 2? 3? and how does this relate to fatigue?
17. In the "Results" section in lines 164-165, the Author write about the results of the health quality scale - "The quality of health scale [15] elicited that the group reported a mean score of 52.69 (SD 27.73), ranging from 15-100" - I think it would be more readable if the author immediately specified what it means that the result is in the range of 15-100.
18. The Author use "Long Covid" and "long COVID" (verse 237). Please check and standardize throughout the text.
19. The Author use "PAC19" or "PAC-19". Please check and harmonize throughout the text, including tables.
20. In the text, the Author use the abbreviation "PFRS", which was first used in the section "Material and methods" (subsection 2.3.7), but the name expansion should be the first time the abbreviation is used.
Author Response
Thank you reviewing the manuscript. I have addressed your comments as follows:
- In the abstract, lines 7, 11, 14, 17, 19, 23: "SAR-2-CoV-2" please correct to "SARS-CoV-2".
This has been corrected throughout
- In the abstract (line 11) and part of the results (line 151), the Author state that the analysis was performed on 26 patients. But then they write about a group of 25 patients (line 157). How many patients were in the 25 or 26 group? Please address this inconsistency.
This has been corrected to 26 participants
- Similarly, in the "Results" section (subsection 3.1.3) lines 172-173. The Author write that the CFS group had a total of 76 patients, but later write that there were 9 men and 68 women, which gives a total of 77 patients - please explain.
This was clarified as follows: One participant did not wish to indicate their sexual orientation.
- In the section "Introduction" (line 32) the Author write "One of the most frequently reported symptoms associated with Long Covid is fatigue." - what is the source of this information? there is no literature.
Citation [2] has been added
- In the "Introduction" section, the Author write "Fatigue is the expected result of over-exertion or lack of sleep and has been defined as an intense subjective sense of tiredness, energy depletion and weakness." - what is the source of this definition? there is no literature.
Citation added
- Is it possible to specify the time period from which the study is? - did all patients meet the WHO definition of long COVID (persistence of symptoms> 12 weeks)?
Time period added: ‘Data were collected between March and August 2021 from individuals identifying as experiencing symptoms of Long Covid (such as fatigue)’.
- The Author write: "Of the 25 respondents, SARS-2-CoV-2 had been formally diagnosed in 17 cases. Twelve by antigen test and 5 by antibody tests." - and in other cases, on what basis was the diagnosis of COVID-19 made?
The following sentence has been added: The remaining participants did not respond to the question.
- There is no detailed description of the criteria that the subjects had to meet in order to be included in the study - to the PAC19 group, and what were the exclusion criteria, e.g. age above ... or below ... etc.
The following criteria have been included in the methodology:
2.1.1 Post-acute SARS-CoV-2
Adults who had been infected with the SARS-CoV-2 virus were recruited from Long Covid support groups via TwitterTM. Inclusion criteria for the study were: (a) adults over the age of 18 years who believed that they had been infected with SARS-CoV-2 (COVID 19) and (b) were experiencing ongoing symptoms (Long Covid). Exclusion criteria for the study were: (a) anyone under 18 years of age who did not believed that they had been infected with SARS-CoV-2 (COVID 19), (b) were not experiencing ongoing symptoms (Long Covid), (c) people who lack mental capacity, may be at risk of losing capacity or have fluctuating capacity, (d) people who suffer from psychiatric or personality disorders, including those conditions in which capacity to consent may fluctuate, (c) people who may have only a basic or elementary knowledge of the English language, (f) people who may socially not be in a position to exercise unfettered informed consent, and (g) any members of staff or students currently at the host university
A total of 31 adults began the questionnaire. Data were excluded if all aspects of the survey were not completed. Of the 29 remaining cases, 3 were excluded from the analysis as they were the only participants who had been hospitalised due to the infection. This precluded them from any meaningful comparative analysis.
2.1.2 Non-fatigue controls
Comparison data from non-fatigue individuals were used as the first comparison group. These anonymised data were collected before the pandemic from volunteers recruited from university staff [dcd study ref]. Inclusion criteria for the study were: (a) adults over the age of 18 years, and (b) were not experiencing symptoms of chronic fatigue. Exclusion criteria for the study were: (a) anyone under 18 years of age who did not believed that they had CFS, (b) people who lack mental capacity, may be at risk of losing capacity or have fluctuating capacity, (c) people who suffer from psychiatric or personality disorders, including those conditions in which capacity to consent may fluctuate, (d) people who may have only a basic or elementary knowledge of the English language, and (e) people who may socially not be in a position to exercise unfettered informed consent. Participants from the study consented to their data being used in research articles.
2.1.3 Chronic Fatigue Syndrome (CFS)
Comparison data from individuals with CFS were used as the second comparison group. These anonymised data were collected before the pandemic from volunteers recruited from the Action for ME support group website [dcd study ref]. Inclusion criteria for the study were: (a) adults over the age of 18 years, and (b) believed they currently had CFS. Exclusion criteria for the study were: (a) anyone under 18 years of age who believed that they had CFS, (b) people who lack mental capacity, may be at risk of losing capacity or have fluctuating capacity, (c) people who suffer from psychiatric or personality disorders, including those conditions in which capacity to consent may fluctuate, (d) people who may have only a basic or elementary knowledge of the English language, and (e) people who may socially not be in a position to exercise unfettered informed consent. Participants from the study consented to their data being used in research articles.
- Similarly, there is a lack of precise information regarding the inclusion and exclusion criteria for the non-fatigue controls group and the CFS group.
See comments in point 8.
- It is not clear whether the "non-fatigue controls" group are people who have contracted COVID-19 who have not noticed fatigue, or people who have not suffered from COVID-19 at all.
See comments in point 8.
- Information on marital status was collected from patients - was there a relationship between the occurrence of fatigue and the marital status of patients?
This is a limitation indicated in the discussion section: we need to collect far more data from those with LC so that further risk factors and predictors of outcome can be examined. This will be addressed by the recruiting from LC outpatients across the UK.
- There is no information anywhere whether the studied patients were taking any medications, eg due to the presence of comorbidities.
As mentioned above, this limitation will be addressed by data collection from LC clinics.
- There is also no information about whether they took any medications during the acute phase of COVID-19 - please add this information as it may be important in the persistence of the observed symptoms.
Information around medicines was included for inpatients. I will clarify this and add this as a limitation.
- In the "Introduction", the Author write: "We have already described the range of cognitive impairments resulting from experimentally induced acute fatigue [3]. This study also highlighted how caffeine can mitigate these effects." - the study collected information related to the lifestyle of patients (diet), e.g. did the respondents use / use caffeinated products? E.t.c.
The question around caffeine was not asked as caffeine is mainly used to reverse the effects of short-term fatigue. It is not (as yet) been suggested that caffeine may help alleviate the fatigue in LC – so will remove the statement.
- In the study, in the "Material and methods" section, the Author write that patients were asked about their vaccination status - but there is no reference to the data obtained - whether in patients after vaccination, fatigue was more or less frequently observed, or whether it was not observed at all. impact? do vaccinations increase / decrease the risk of fatigue after COVID-19?
The following information has been added: ‘Eighteen participants reported receiving a first dose of a SARS-CoV-2 vaccine, 8 had not received vaccine. Seven received the AstraZenica/Oxford vaccine and 11 received the Pitzer-BioNTech vaccine. Vaccination dates were between December 2020 and June 2021. Eleven participants had received a second vaccine dose. Two had received the AstraZenica/Oxford vaccine and 9 the Pitzer-BioNTech vaccine. Second vaccines were received between February 2021 and July 2021. In total, 11 participants had received 2 vaccine doses.’
- How many patients in this group have completed the full vaccination course? How many was 1? 2? 3? and how does this relate to fatigue?
See point 15
- In the "Results" section in lines 164-165, the Author write about the results of the health quality scale - "The quality of health scale [15] elicited that the group reported a mean score of 52.69 (SD 27.73), ranging from 15-100" - I think it would be more readable if the author immediately specified what it means that the result is in the range of 15-100.
These has been amended to read: The quality of health scale [15] data ranged from 15-100 and elicited that the group reported a mean score of 52.69 (SD 27.73), ranging from 15-100"
- The Author use "Long Covid" and "long COVID" (verse 237). Please check and standardize throughout the text.
The term Long Covid has now been replaced throughout
- The Author use "PAC19" or "PAC-19". Please check and harmonize throughout the text, including tables.
The term PAC-19 has now been replaced throughout
- In the text, the Author use the abbreviation "PFRS", which was first used in the section "Material and methods" (subsection 2.3.7), but the name expansion should be the first time the abbreviation is used.
PFRS has been added in brackets after the first mention of the scale (2.3.7)
Round 2
Reviewer 1 Report
The author has addressed all points raised by me, therefore the article now is typical for publishing.